

# Invited perspectives: A research agenda towards Disaster Risk Management pathways in multi-risk assessment

Philip J. Ward[1], James Daniell[2], Melanie Duncan[3], Anna Dunne[4], Cédric Hananel[4], Stefan Hochrainer-Stigler[5], Annegien Tijssen[6], Silvia Torresan[7], Roxana Ciurean[3], Joel C. Gill[3], Jana Sillmann[8], Anaïs Couasnon[1], Elco Koks[1], Noemi Padrón-Fumero[9], Sharon Tatman[6], Marianne Tronstad Lund[8], Adewole Adesiyun[10], Jeroen, C.J.H. Aerts[1,6], Alexander Alabaster[11], Bernard Bulder[12], Carlos Campillo Torres[13], Andrea Critto[7], Raúl Hernández Martín[9], Marta Machado[14], Jaroslav Mysiak[7], Rene Orth[15], Irene Palomino[13], Eva-Cristina Petrescu[16], Markus Reichstein[16], Timothy Tiggeloven[1], Anne F. van Loon[1], Hung Vuong Pham[17], Marleen C. de Ruiter[1]

[1]Institute for Environmental Studies (IVM), Vrije Universiteit Amsterdam, De Boelelaan 1111, 1081 HV Amsterdam, The Netherlands
[2]Risklayer, Haid und Neu Str. 7, 76131 Karlsruhe, Germany
[3]British Geological Survey, The Lyell Centre, Edinburgh, EH14 4BA, United Kingdom & Nicker Hill, Keyworth, Notts, NG12 5GG, United Kingdom
[4]Arctik, Avenue de Broqueville 12, 1150 Woluwe-Saint-Pierre, Belgium
[5]IIASA-International Institute for Applied Systems Analysis, Schlossplatz 1, 2361 Laxenburg, Austria
[6]Deltares, Boussinesqweg 1, 2629 HV Delft, The Netherlands
[7]Centro Euro-Mediterraneo sui Cambiamenti Climatici, VIA A Imperatore 16, 73100 Lecce, Italy
[8]CICERO, Gaustadallèen 21, 0349 Oslo, Norway
[9]Universidad de la Laguna, Molinos de Agua, 38071 La Laguna, Tenerife, Spain
[10]FEHRL, Boulevard de la Woluwe 42, 1200 Brussels, Belgium
[11]AON, 8 Devonshire Square, London EC2M 4PL, United Kingdom
[12]TNO, Anna van Buerenplein 1, 2595 DA The Hague, The Netherlands
[13]CICYTEX, Autovia A5 KM 372 Finca La Orden Guad, 06187 Lobon Badajoz, Spain
[14]HOTREC, Dautzenberg 36, 1050 Brussels, Belgium,
[15]Department of Biogeochemical Integration, Max Planck Institute for Biogeochemistry, Hans-Knöll-Straße 10, D-07745 Jena, Germany
[16]Bucharest University of Economic Studies (ASE Bucharest), 6 Piata Romana, 010374 Bucharest, Romania
[17]Dept. of Environmental Sciences Informatics and Statistics, University Ca' Foscari Venice, Via delle Industrie 21/8, c/o INCA - VEGAPARK, 30175 Marghera-Venice, Italy

*Correspondence to: Philip J. Ward (philip.ward@vu.nl)*

**Abstract.** Whilst the last decades have seen a clear shift in emphasis from managing natural hazards to managing risk, the majority of natural hazard risk research still focuses on single hazards. Internationally, there are calls for more attention for multi-hazards and multi-risks. Within the European Union (EU), the concepts of multi-hazard and multi-risk assessment and management have taken centre stage in recent years. In this perspective paper, we outline several key developments in multi-hazard and multi-risk research in the last decade, with a particular focus on the EU. We present challenges for multi-risk management as outlined in several research projects and papers. We then present a research agenda for addressing these challenges. We argue for an approach that addresses multi-hazard, multi-risk management through the lens of sustainability challenges that cut across sectors, regions, and hazards. In this approach, the starting point is a specific sustainability challenge,



40    rather than an individual hazard or sector, and trade-offs and synergies are examined across sectors, regions, and hazards. We argue for in-depth case studies in which various approaches for multi-hazard and multi-risk management are co-developed and tested in practice. Finally, we present a new pan-European research project in which our proposed research agenda will be implemented, with the goal of enabling stakeholders to develop forward-looking disaster risk management pathways that assess trade-offs and synergies of various strategies across sectors, hazards, and scales.

## 1 From managing disasters to managing risk: a potted history

An enduring story in classical Greek mythology is the battle between the Greek hero Hercules and the river god Achelous. Hercules defeats Achelous, who has taken the form of a bull, by wrenching off one of his horns. The horn becomes the Cornucopia, or Horn of Plenty, a symbol of abundance depicted as a large horn overflowing with produce. One of many interpretations of this myth suggests that Hercules' victory represents engineering operations, including channels, embankments, and dams, by which rivers were tamed from the vagaries of flooding to create a fertile tract of land for cultivation (Bengal, 1847). Therefore, the story can be interpreted as an early example of humans' efforts to master our natural environment. However, classic folklore and mythology also abound with examples of nature punishing humans for their treatment of the planet. Indeed, the great Roman author, philosopher, and geographer Pliny the Elder wrote in his classic Naturalis Historia from ~79 AD: "We trace out all the veins of the earth, and yet, living upon it, undermined as it is beneath our feet, are astonished that it should occasionally cleave asunder or tremble: as though, forsooth, these signs could be any other than expressions of the indignation felt by our sacred parent!" Pliny the Elder (cited in Bostock & Riley, 1857). From the Enlightenment period, the scientific study of natural hazards moved from viewing disasters as punishments by nature or "acts of God", towards trying to explain and understand natural cause-effect relationships.

Clearly, then, there is long-standing understanding and philosophical dialogue about the role of humankind in aggravating or mitigating the impacts of natural hazards. Since the start of the industrial era, up until the 1970s, the focus of most research and practice was on managing what were long referred to as natural disasters (Burton, 2005; Peduzzi, 2019). In this worldview, humans use their know-how to design and implement measures to keep hazards at bay. Take for example Hercules' victory from the opening passage of this paper.

As our world becomes ever more populated and human settlements continue to expand at alarming rates, the impacts of natural hazards have increased sharply over the last half century (Poljanšek et al., 2017). Indeed, globalisation and the concentration of population around large cities increases exposure to climate and other natural hazard risks. Globally, over the last 20 years natural hazards have caused an estimated 931,000 direct fatalities (excluding heatwave deaths) and over €3.87 trillion (2021 inflation adjusted) in economic losses over the last 20 years (CATDAT, 2021), and have affected more than 200 million people per year on average (CRED, 2021). Since the 1970s, the idea that disasters are also a human construct became (slowly) accepted (Peduzzi, 2019), and is now well embedded in the literature (Kelman et al., 2016).



Accordingly, recent decades have seen a move from managing hazards to managing risks, as documented in several works. This perspective paper is not the place to review this journey: excellent histories are provided in several scholarly works, including Zentel and Glade (2013), Tozier de la Poterie and Baudoin (2015), Peduzzi (2019), Aronsson-Storrier (2020). Suffice to say that an important point in this journey was the ratification of the Hyogo Framework for Action 2005-2015 (HFA), an

outcome of the World Conference for Disaster Reduction, held in Kobe in 2005. This followed the International Decade for Natural Disaster Reduction 1990-2000, and the subsequent creation of the United Nations International Strategy for Disaster Reduction (UNISDR) in 1999 (which was subsequently renamed United Nations Office for Disaster Risk Reduction, or UNDRR). The HFA is likely the most significant international document popularising the notion of disaster risk reduction, reflecting a stronger focus on risk preparedness and prevention as opposed to the emphasis on response and recovery during

the previous decades. The HFA was also the first international framework describing the detailed processes needed to reduce disaster risks across different geographical scales and sectors (Tozier de la Poterie and Baudoin, 2015).

The developments outlined above have led to a growing understanding and body of research on disaster risk. The vast majority of that research has focused on single hazards. Internationally, there is an ongoing call for more attention towards multi-hazards and multi-risks. Within the European Union (EU), the concepts of multi-hazard and multi-risk assessment and management

have taken centre stage in recent years. In this perspective paper, we outline several key developments in multi-hazard and multi-risk research in the last decade, with a particular focus on the EU (Section 2). We also present challenges for multi-risk management (Section 3) as outlined in several research projects and papers, with an emphasis on Europe. Finally, we present a research agenda that will be implemented in an upcoming project for addressing these challenges.

## 2 The whole is greater than the sum of its parts: from risk to multi-risk

The interaction of different hazards can lead to an impact that is greater than the sum of the single hazard effects (Kappes et al., 2012; Terzi et al., 2019). Whilst Hewitt and Burton (1971) noted a half century ago the need to shift natural hazards research from a single-hazard approach towards a systematic cross-hazard approach, concerted calls for multi-hazard and multi-risk approaches only date back to the early 1990s. The Agenda 21 for sustainable development (UNCED, 1992) called for a "*complete multi-hazard research*" approach to human settlement planning and disaster risk (Scolobig et al., 2017). This notion

was taken up in the HFA, and takes even more centre-stage in the Sendai Framework for Disaster Risk Reduction 2015-2030 (Sendai Framework) (UNDRR, 2015). The Sendai Framework calls to "*promote investments in innovation and technology development in long-term, multi-hazard and solution-driven research in disaster risk management*". In 2017, the UNDRR terminology first included a definition of multi-hazard, meaning "*(1) the selection of multiple major hazards that the country faces, and (2) the specific contexts where hazardous events may occur simultaneously, cascadingly or cumulatively over time,*

*and taking into account the potential interrelated effects.*"

Several EU policies, strategies, and frameworks advocate for a multi-hazard, multi-risk, and multi-sector approach. For example, the Internal Security Strategy (SEC (2010) 1626 Final) (which evolved into the European Agenda on Security)





advocates for an "*all-hazard approach to threat and risk assessment*"; the EU Community Framework on Disaster Prevention point 22 underlines "*the usefulness of a multi-hazard approach*" in Regulation 1313/2013/EU; and the European Disaster Risk
Reduction Strategy (COM 2008, 130) states the need for comprehensive approaches to disaster management (Poljanšek et al., 2017).

The move towards a multi-hazard and multi-risk approach is reflected in the research agenda of the European Union, where the topic has been within its Framework Programmes (FPs) since FP4. Several major EU-funded projects are listed in Table 1. The move in science towards this approach is reflected in the creation of a subdivision on multi-hazard risk within the
European Geosciences Union in 2019. The theme of multi-hazard approaches has been central to a series of conferences between the European Geosciences Union and the Asia-Oceania Geosciences Society, since 2017, on 'New Dimensions for Natural Hazards in Asia'.

**Table 1: Selected major European-funded multi-hazard and multi-risk projects**

| Project | Name | Period |
|---|---|---|
| TIGRA | The Integrated Geological Risk Assessment | 1996-1997 |
| TEMRAP | The European Multi-Hazard Risk Assessment Project | 1998-2000 |
| Na.R.As | Natural Risks Assessment harmonisation of procedures, quantification and information | 2004-2006 |
| ARMONIA | Applied Multi-Risk Mapping Of Natural Hazards for Impact Assessment | 2004-2007 |
| MATRIX | New Multi-Hazard and Multi-Risk assessment methods for Europe | 2010-2013 |
| ENHANCE | Enhancing Risk Management Partnerships for Catastrophic Natural Disasters in Europe | 2012-2016 |
| STREST | Harmonized Approach to Stress Tests for Critical Infrastructures against Low-Probability High-Impact Natural Hazards | 2013-2016 |
| ASAMPSA_E | Advanced Safety Assessment Methodologies | 2013-2016 |
| FORTRESS | Foresight Tools for Responding to Cascading Effects in a Crisis, 2014-2017 | 2014-2017 |
| NARSIS | New Approach to Reactor Safety Improvements | 2017-2022 |
| ARISTOTLE-eENHSP | All Risk Integrated System TOwards Trans-boundary hoListic Early-warning - enhanced European Natural Hazards Scientific Partnership | 2020-2023 |

The industry and policy-driven demand for, and science-driven supply of, multi-risk knowledge has led to a proliferation of myriad approaches for multi-risk assessment. Several reviews of these approaches can be found in various reports and papers,



including Kappes et al. (2012), Gill and Malamud (2014), Poljanšek et al. (2017), Scolobig et al. (2017), Ciurean et al. (2018), and Tilloy et al. (2019). In particular, Ciurean et al. (2018) divide these approaches into several main classes: narrative descriptions, hazard wheels, hazard matrices, network diagrams, hazard maps, hazard/risk indices, systems-based or physical modelling, and probabilistic and statistical approaches.

Moreover, recent years have seen the development of several networks focusing on various aspects of multi-hazard and multi-risk assessment and management. UNDRR launched the Global Risk Assessment Framework, with a Working Group specifically dedicated to *Fostering Systems Thinking*, in which risk is addressed through a multi-hazard and multi-sector lens. Major research programmes - Future Earth, IRDR, WCRP, WWRP - have formed an interdisciplinary network centred on systemic risk (Risk-KAN). Within the climate community, several networks have formed around the concept of compound climate extremes, including the EC Cost Action DAMOCLES (Understanding and Modelling Compound Climate and Weather Events) and the Risk-KAN Working Groups on Compound Events and Impacts, and on Early Warnings for Systemic Risk.

## 3 Challenges for multi-risk management

Notwithstanding the many advances made in the last decades, multi-risks are still not mainstreamed in disaster risk management (DRM) (Poljanšek et al., 2017, Zscheischler et al., 2018). Indeed, most research and policy still addresses risk from a single-hazard, single-sector, perspective. This presents challenges for addressing real-world challenges faced by risk managers and other decision-makers. Firstly, multiple hazards can interrelate, and this can contribute to changes in risk. For example, an earthquake could trigger a landslide; dry conditions could amplify the likelihood of forest fires; a combination of rainfall and storm surge could cause compound flooding; or a region could face several consecutive hazards, with changes in exposure and/or vulnerability between these. How can risk be better managed by considering these interrelated effects? Secondly, DRM measures taken to reduce risk from one hazard may increase risk from another hazard. For example, wood-frame buildings may perform well in earthquakes, but could sustain high damages during flooding (and fires). How can we better account for these dynamic feedbacks between risk drivers? Thirdly, these interrelated effects have impacts across sectors and regions. For example, there are trade-offs and synergies between maintaining the sustainable use of our land and marine regions while meeting increasing demand for sustainable energy and food and reducing natural hazard risks. How can we account for these trade-offs and synergies across sectors, regions, and hazards?

The aforementioned challenges exist within the context of an increasingly interconnected world, increased pressure for space, and climate change, in which the magnitude and frequency of single and multi-hazards are changing at an unprecedented rate (Vogel et al., 2019). A paradigm shift is needed to successfully address these complex questions and challenges, in which science and practice move from a single-hazard, single-sector risk perspective towards a multi-risk, multi-sector, systemic approach. This approach should embrace risk-aware sustainable development, which acknowledges that sustainable development goals are endangered by multi-hazards and systemic risk, but at the same time can contribute strongly to systemic resilience (Reichstein et al., 2021). The COVID19 situation lays bare the interconnections between sectors, regions, and



hazards, as its impacts propagated geographically and across sectors (Lopez Prol 2020, OECD 2021) which highlights the need

for a more systemic approach to reducing risk. Several concrete challenges hindering the movement towards this approach relate to existing knowledge gaps in multi-hazard risk assessment and management, such as those described in recent reviews by Ciurean et al. (2018) and Tilloy et al. (2019). Here, we give a brief overview of several of these key challenges.

**There is a diverse language on multi-hazard risk and a lack of overview of existing methods and tools.** Existing reviews of multi-hazard approaches have shown diverse and conflicting language used to characterise multi-hazard risk (Kappes et al.,

2012; Gallina et al., 2016; Ciurean et al., 2018; Tilloy et al., 2019). The inclusion of a definition of 'multi-hazard' in the UNDRR terminology (UNDRR, 2016) may help bring clarity to this term, but there is still a lack of consensus within and between research, industry, and policy communities around the varied multi-risk terminology. Moreover, whilst there are myriad qualitative and quantitative methods to support multi-hazard risk assessment (Sperotto et al., 2017; Ciurean et al., 2018; Terzi et al., 2019; Tilloy et al., 2019), they are highly dispersed through different scientific communities, often across multiple

languages, disciplines, and publication types (Ciurean et al., 2018).

**We lack a clear framework and guidelines for multi-risk assessment and management.** Conventional risk assessment and management usually consider different hazards and risks as independent from each other (Scolobig et al., 2017; De Ruiter et al., 2020). In this classic approach, individual hazards and sectors are the point of departure. In the case of multi-hazard situations, various methodologies are now suggested that focus on specific aspects, including compound events, cascading

effects, or systemic risks (Tilloy et al., 2019). These different aspects are usually treated separately within such assessments. The separation of the analysis of multi-risk into different compartments is not a coincidence, indeed multi-risk assessment and management is complex. Whilst a framework for multi-risk governance has been developed in the MATRIX project (Scolobig et al., 2017), an overall framework for multi-risk assessment and management is missing. Moreover, according to interviews conducted with risk managers within MATRIX, many of them miss criteria or guidelines that would help them to carry out a

multi-risk assessment. Many of them mentioned that current multi-risk assessment methods required a large degree of expertise and that many of the available tools are not user-friendly (Poljanšek et al., 2017).

**Dynamic feedbacks between hazard, exposure, and vulnerability are poorly represented (Gill and Malamud, 2014, 2016).** Databases such as the GED4ALL Global Exposure Database for Multi-Hazard Risk Analysis (Silva et al., 2018) and the multiple-hazard data scheme as proposed by Murnane et al. (2019) allow for cross-hazard comparisons of risk, but do not

account for dynamics and feedback loops between the different components of risk. Risk models have been developed to assess changes in risk in the past and future due to changes in hazard, exposure, and (to a much lesser extent) vulnerability (Ward et al., 2020a). Also, studies have examined long-term trends in reported losses and damages (Bouwer, 2018; Paprotny et al., 2018). However, they examine long-term trends, assuming no interactions between risk drivers. Changes in exposure and/or vulnerability can influence the occurrence of multi-hazard events. For example, change in agricultural practice change,

vegetation removal, surface and subsurface construction, quarrying, and so forth can trigger natural hazards or amplify multi-hazard interrelationships (Gill et al., 2021). Conversely, progression through a multi-hazard event can result in changes to exposure and/or vulnerability (De Ruiter et al., 2020), such as changes to exposure when community members are relocated





after a volcanic eruption. New dynamic modelling approaches need to be developed which can tackle indirect and emergent risks which materialize through the interaction of physical and ecological systems and several societal actors (Reichstein et al., 2020, 2021)

**There is a distinct lack of in-depth case-studies on multi-risk assessment and management.** Recent years have seen an increase in communities working on different aspects of multi-hazards (e.g. triggering relationships (Gill et al., 2020), compound events (Zscheischler et al., 2018), consecutive events (De Ruiter et al., 2020)). However, most past multi-risk case studies are still limited to one or two specific hazards at a given site (Ciurean et al., 2018), whilst real-life situations involve multiple hazard types and interrelated effects across various spatial and temporal scales (Tilloy et al., 2019; Ward et al. 2020b). Theoretical multi-risk approaches are often based on hypothetical data and focus on simulating hazard time-series without addressing the impacts of spatiotemporal interactions between hazards (Mignan et al., 2014). When multi-hazard scenarios have been used to assess risk, this has focused on local scale impacts and specific sectors (Tilloy et al., 2019).

**Many past multi-risk projects & software focus on multiple single hazards, and tend not to assess future scenarios (Gallina et al., 2016).** Large-scale multi-hazard studies have primarily assessed impacts of each hazard individually (Koks et al., 2019), without considering interrelations. Only at local scales have complex impacts resulting from multi-hazard interactions been assessed. For example, a recent case study for the north of the Netherlands examined potential structural damage to masonry housing due to sequences of earthquakes and earthquake-triggered floods (Korswagen et al., 2019). However, such approaches are rare. In the context of future scenarios, different development trajectories may change multi-hazard risk. For example, in the context of urbanisation and urban expansion, future or 'potential risk' depends on unbuilt infrastructure, unknown socio-economic characteristics, and unmade decisions (Galasso et al., 2021).

**Only few studies examine differences in the effectiveness of DRM measures between single and multi-hazard scenarios and between sectors.** From an engineering perspective, some knowledge exists about the (a)synergies of different building practices (Zaghi et al., 2016). The Building Back Better (Hallegatte et al., 2018) research focuses mainly on critical infrastructure (e.g. bridges, schools, hospitals) (Li et al., 2012). Nonetheless, trade-offs and synergies of DRM measures are not commonly quantified in risk assessments, which can lead to maladaptation (Liu et al., 2014; De Ruiter et al., 2021a).

**Development plans should use time horizons of several decades, with associated deep uncertainty.** Long-term plans and strategic decisions need to be based on highly uncertain risk information (Peduzzi, 2019). Ignoring uncertainty could mean that we limit our ability to adapt and can result in missed chances and opportunities. Multi-hazard and cross-sectoral trade-offs and synergies increase the complexity of these strategic planning challenges. The Dynamic Adaptation Policy Pathways (DAPP) approach has been developed to support decision-making under deep uncertainty (Haasnoot et al., 2019), and successfully applied in several single-hazard decision-making contexts related to climate change (e.g. Thames Estuary 2100 Project, UK (Ranger et al., 2013); Delta Program, Netherlands (Bloemen et al., 2018)). However, the approach lacks guidance for use in a multi-risk setting. This deep understanding of multi-hazards and their uncertainties is also key for shaping economic systems towards sustainability and climate change resilience. As multi-hazard risks are systematically weakly-internalised in





long term asset management and valuation, this can lead to insurance gaps and potentially misleading decisions in capital markets and government planning.

**There is a lack of decision makers/institutions dealing with interconnected multi-risks across sectors, hazards and regions.** Transboundary risk assessment and management for multi-hazards across countries (e.g. the 2002 flood events in Eastern Europe or the heat wave in 2003) is still lacking but very much needed in our ever increasingly interconnected world. Spillover effects from one sector in one country to another sector in another country can hardly be managed if there is no decision maker/or institution responsible for the assessment and the management of such transboundary risks. As has been seen with the financial crisis in 2007/08, even in the case that individuals behave rationally in their own way (e.g. making profit) the systemic risk that they were creating together led to the near collapse of the system. The interconnectedness of multi-hazards and multi-risks need to be explicitly taken into account, both from the individual and system (e.g. country/transboundary) perspective, may it be in relation to the economic or social or ecological system at hand, so that individual DRM measures do not produce systemic risks on other scales/sectors in the future.

## 4 A research agenda: sustainable DRM pathways through challenge-based research in real-world settings

The challenges outlined above demonstrate that there is a long way to go before the much lauded multi-risk approach is mainstreamed in decision making. We argue for an approach that addresses multi-hazard, multi-risk management through the lens of sustainability challenges that cut across sectors, regions, and hazards. In this approach, the starting point is a specific sustainability challenge, rather than an individual hazard or sector, and trade-offs and synergies are examined across sectors, regions, and hazards. Where typical risk assessments try to address questions such as '*What is the risk to sector X of hazard Y in region Z and what DRM measures can be taken to reduce that risk?*', this approach requires questions such as '*What DRM pathways are available to develop a sustainable future in region Z that account for trade-offs, synergies, and interactions across relevant hazards and sectors, and consider interregional linkages?*'. This requires a much more explicit link between the goals of the Sendai Framework and those associated with other policy goals and frameworks such as the Sustainable Development Goals and the Paris Agreement on Climate Change.

We present a research agenda to help us move towards this approach. This agenda will be implemented in the EU Horizon 2020 project MYRIAD-EU (**M**ulti-hazard and s**Y**stemic framework for enhancing **R**isk-**I**nformed m**A**nagement and **D**ecision-making in the **EU**). The overall aim of MYRIAD-EU is that by its completion, policy-makers, decision-makers, and practitioners will be able to develop forward-looking DRM pathways that assess trade-offs and synergies of various strategies across sectors, hazards, and scales. The research agenda, which explicitly addresses the challenges mentioned in Section 3, is presented below.

**Establishing a set of common multi-hazard, multi-risk concepts, definitions, and indicators.** Improving consensus around definitions and providing a clear overview of existing indicators, methods, and tools would help to improve communication and ensure that multi-risk management approaches meet the expectations of the Sendai Framework. Indeed, recommendation



3 of the recent UNDRR Technical Working Group on the Hazard Definition Classification Review is '*Engaging with users and sectors for greater alignment and consistency of hazard definitions*' (UNDRR, 2020). To address this part of our agenda,

we will develop a handbook of multi-hazard, multi-risk concepts, definitions, and indicators, a WIKI-style online crowdsourcing platform of examples of qualitative and quantitative multi-hazard, multi-risk methods, models, and tools, and an overview of existing policies relating to multi-hazard risk management at diverse scales.

**Co-developing a harmonised framework for multi-hazard, multi-sector, systemic risk management**. We propose a multi-risk framework that addresses future sustainability challenges (e.g. spatial planning on land or in the sea), rather than the classic

approach where individual hazards/sectors are the point of departure. The framework will be made concrete by a set of practical guidelines for carrying out a multi-hazard risk assessment, formalised in documented guidance protocols. Moreover, we see the need for a user-friendly web-based dashboard that provides access to state-of-the-art multi-risk products and services from across the multi-risk community. This reflects our conviction that there is no one-size-fits-all solution, and that continuous learning across projects and disciplines is needed to break the silos in which natural hazard risk science operates.

**Increasing understanding of dynamic feedbacks between risk drivers.** We propose an online database of empirical evidence of multi-hazard risk dynamics, which can be used to develop functions to represent these dynamics in multi-risk models. By modelling exposure and vulnerability profiles dynamically, DRM actions can be assessed that consider where development occurs and how this can be changed to reduce future losses. This can support demonstrating the effectiveness of land use planning and risk-sensitive developments as a DRM action. Moreover, serious games, such as 'Breaking the Silos'

(De Ruiter et al., 2021b), can help support stakeholders in understanding the complexities of feedbacks between different DRM measures in a multi-risk setting.

**Developing future scenarios of plausible multi-hazard scenarios.** Datasets and time-series of current and future scenarios of individual hazards have been developed in many past studies. We see the need for user-friendly software to allow users to generate realistic multi-hazard stochastic event sets at subnational to European scales that include different hazard

interrelationships (triggering, amplifying, compound, consecutive). For example, this could be achieved by combining existing single-hazard data and scenarios with state-of-the-art statistical methods (e.g. multivariate methods like copulas, Markov chains, Bayesian Belief Networks) (e.g. Schäfer & Wenzel, 2017; Ward et al., 2018; Tilloy et al., 2019).

**Assessing the effectiveness of DRM measures across multi-hazard scenarios and between sectors.** The pathways approach to adaptation planning has been applied in many single-hazard contexts, and proven its usefulness for planning under deep

uncertainty. Extending the current DAPP approach to be fit for use in a multi-risk setting would ensure that the decision context and processes that govern multi-hazards and risks, and their compound and cascading effects, are considered throughout the whole process, from problem setting, to risk assessment, and finally decision making. This will allow the assessment of whether (and what) different decisions would be taken when adopting a multi-hazard and multi-sector lens, in which trade-offs and synergies between hazards, sectors, regions, and decision and policy goals, compared to a single-hazard and single-sector lens.

**Testing of approaches in in-depth case studies**. We see the need for in-depth case-studies in which our framework and the various approaches for multi-risk assessment and management are tested in practice. The MYRIAD-EU approach aims to





achieve this by co-developing the framework, and products and services to operationalise the framework, with stakeholders in five multi-scale Pilots: North Sea, Canary Islands, Scandinavia, Danube, and Veneto (Figure 1). The Pilots focus on forward-looking DRM solutions to real-world sustainability challenges, such as: How can spatial planning at the interface of the land

and sea environments be optimised in the face of increasing and interrelated risk?; How can we maintain healthy ecosystems while meeting increasing demand for energy, food, and ecosystem services?; How can we increase resilience to multiple disasters that impact interconnected countries with strong macro-economic relations? They assess a spread of different hazards (meteorological, geological, biological, hydrological) as well as hazard interrelationships (triggering, amplifying, consecutive, and compound). Each Pilot focuses on (interlinkages between) several key economic sectors: infrastructure & transport, food

& agriculture, ecosystems & forestry, energy, finance, and tourism. For each Pilot, we examine multi-hazard risk within the Pilot region, as well as indirect, cross-sectoral, and interregional risks throughout the rest of Europe.

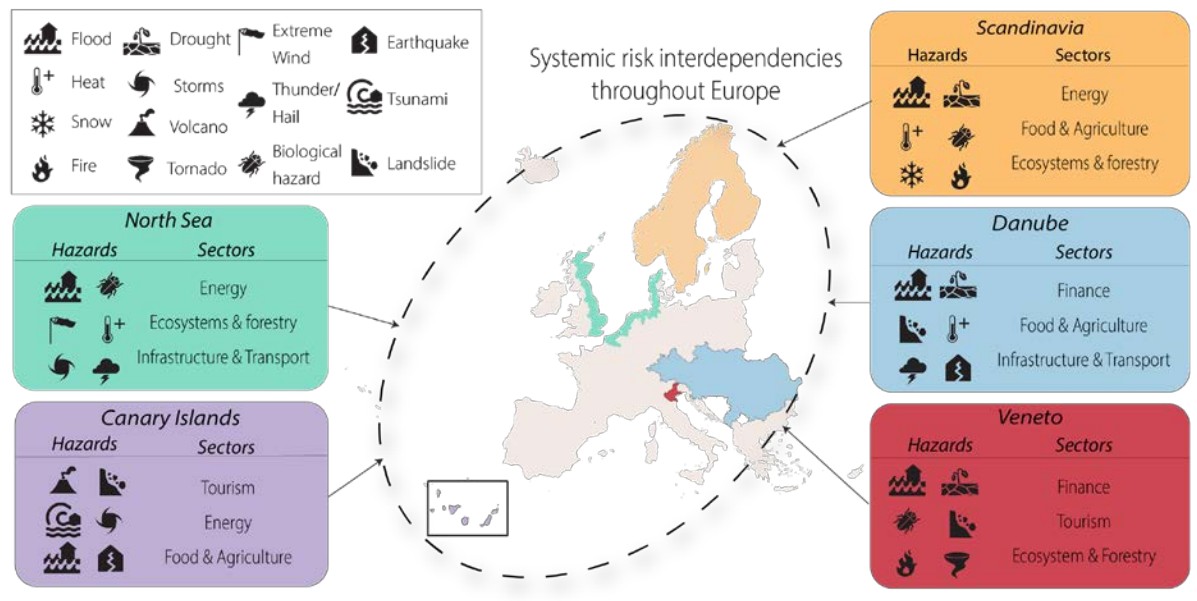

**Figure 1: Overview of sustainability challenges to be addressed in MYRIAD-EU project**



Through the MYRIAD-EU project, we intend to contribute to the proposed research agenda. From our perspective, an indicator
of success will be if the policy-makers, decision-makers, and practitioners with whom we collaborate in the five pilots have
been able to develop forward-looking DRM pathways for their region that assess trade-offs and synergies of various strategies
across sectors, hazards, and scales. The multi-risk framework, methods developed, and knowledge generated should also be
suitable for use in case studies throughout Europe and elsewhere. Another indicator of success is therefore their uptake within
wider DRM projects, networks, and dialogues. Of course, this research agenda is no panacea. Just as our contextualisation of
disasters, hazards, and risks has evolved throughout history, DRM practice must continue to evolve as society's understanding
of risk improves and the nature of the risks it faces changes.

**Author contribution**

PJW coordinated and led the writing, in close collaboration with MdR. All authors contributed to the conceptualisation of the
paper, discussions on the content, and contributing text and ideas.

**Competing interests**

The authors declare that they have no conflict of interest.

**Acknowledgements**

MYRIAD-EU received funding from the European Union's Horizon 2020 research and innovation programme under grant
agreement No 101003276. The work reflects only the author's view and that the Agency is not responsible for any use that
may be made of the information it contains. PJW, AC, and TT also received funding from the Dutch Research Council (NWO)
in the form of a VIDI grant (grant no. 016.161.324). Melanie Duncan, Roxana Ciurean and Joel Gill publish with permission
of the Executive Director, British Geological Survey (UKRI).

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
