# Peer review of "Invited perspectives: A research agenda towards Disaster Risk Management pathways in multi-(hazard)-risk assessment"

_Natural Hazards and Earth System Sciences, 2021_

## Author Response (AR1)

**Reply to reviewers: NHESS-2021-326**

We would like to thank the editor and two reviewers for the time taken to review and process our manuscript. We respond to the reviewer remarks below. The reviewer remarks are shown in *italics*. We have also checked the reference list as requested by the editor. With regards the specific comment about the Terzi reference, we are not sure what is meant, since the reference already includes the title of the contribution.

**Reply to David N. Bresch**

*The paper is well devised and written. Nice to see the paper start from Greek mythology and good to see the paper swiftly becomes more tangible, indeed. Good also to take a short cut to the core, rather than reviewing the scholarly journey from managing hazards to risks. Laudible to recount Hyogo and Sendai (that part might be a bit on the longish side), too. The paper then comments on many of the known challenges in multi-risk assessments and the lack of case studies at least in the academic literature. Sad to note that there are quite some in reports etc., especially in the context of local or regional climate adaptation. The research agenda presented does clearly state the cornerstones of such an endeavour, yet the proof will be in the (open-source and -access) implementation and its transferability, which will very much depend on the interoperability of (sub)models and modules. Hence some emphasis might have been put on interoperability rather than a fully unified framework which might take a long time to emerge and even more so to be readily applicable. The approach from the problem side, i.e. embracing sustainability challenges, looks promising - yet framing the problem from a resiliency angle (i.e. the strengthening thereof) might proof to be very valuable, too.*

We thank David Bresch for the time taken to review our manuscript and for the positive feedback and interesting thoughts. We agree that interoperability is a very important issue, especially for developing models and tools within this realm. Within MYRIAD-EU, the "framework" itself is intended to be broader, providing a set of concrete guidelines for designing multi-risk reduction pathways. It is therefore not a model in itself. In the revised manuscript we clarify this point by adding: "The framework will be co-developed within the project between the consortium and our stakeholders in the pilot regions, which will involve an iterative process of framework development, testing, feedback, and updating. The framework is intended to provide a set of practical guidelines for carrying out a multi-(hazard)-risk assessment. We explicitly do not aim to develop a unified method or model for navigating the framework, as it is our conviction that there is no one-size-fits-all model for addressing multi-(hazard)-risk management, and that continuous learning across projects and disciplines is needed to break the silos in which natural hazard risk science operates. Instead, we see the need for a user-friendly web-based dashboard that provides access to a myriad of state-of-the-art multi-(hazard)-risk products and services from across the multi-(hazard)-risk community." We will, as stated in the paper, develop a software package for generating multi-hazard stochastic event sets based on pre-computed inputs in certain formats from either independent hazard models, or through other software packages and existing stochastic hazard models. It is our goal to extend the state-of-art in the multi-hazard sphere, and plan to integrate it with the existing independent hazard solutions already available around the world along the lines of compatibility and interoperability. We added the following sentence to clarify the importance of

interoperability, open-access, and so forth: "Such a software package should be open-source and open-access, and allow for interoperability with other software packages, datasets, and models."

**Reply to Kai Kornhuber**

*The article by Ward et al. provides an interesting perspective on how to respond and manage multi-hazard and their increasing risks in light of global changes such as globalization and climate changes. It offers a comprehensive overview of the historic developments in disaster risk management, summarizes challenges of traditional approaches and provides a suggested way forward, centred around sustainability. I particularly enjoyed the introductory references and remarks on ancient history and would very much welcome to see this paper published as it offers a helpful reference, a nicely written entry point to the general topic and some new important concepts. I therefore have only some minor suggestions and very much hope the authors find them useful.*

We thank Kai Kornhuber for the time taken to review our manuscript and for the positive feedback. We respond to the individual comments below.

*As this is a perspective that highlights the need to unify language for progress it would be very helpful if some important terms such as disasters, hazards, risks and their multi-counterparts ('multiple-disasters', 'multi-hazards', 'multi-risks' ) are defined early on and then used consistently throughout. Currently they are used synonymously (e.g. Section 2 is headed risk to multi-risks but talks mostly about multi-hazards, later multiple disasters is used as well. e.g.l.285), but are they really the same? If multi-hazards lead to amplified risk (a probability?) of severe impacts, what is a 'multi-risk' etc.?*

Thank you very much for picking us up on this. This is indeed an excellent demonstration of the first challenge listed in the paper, namely "There is a diverse language on multi-(hazard)-risk and a lack of overview of existing methods and tools". During the first 6 months of the MYRIAD-EU project, we have been carrying out a review of the literature in order to try to establish a set of common definitions, at least within the project. Based on this review, we have now reviewed the entire manuscript to make sure that we use consistent terminology. To that end, we have added the following to section 1:

"It should be noted that there are many different terminologies used in this rapidly evolving field. Whilst this paper is not the place to review this terminology, Table 1 shows several key definitions of the terms multi-hazard, multi-hazard risk, and multi-risk. For the sake of simplicity, in this paper we use the term multi-(hazard)-risk when referring to all of these different aspects collectively. For terminology related to risk in general, we follow UNDRR (2016)."

We also added the table of key definitions also shown below. We now use these terminology consistently within the paper, and have adjusted the paper accordingly. We prefer not to repeat definitions of aspects such as risk, hazard, exposure, and vulnerability (etc) without the multi-aspect, but do state to the reader that "For terminology related to risk in general, we follow UNDRR (2016)."

Table 1: Key definitions related to multi-(hazard)-risk

| Term | Definition | Source |
|------|-----------|--------|
| Multi-hazard | The selection of multiple major hazards that the country faces, and the specific contexts where hazardous events may occur simultaneously, cascadingly or cumulatively over time, and taking into account the potential interrelated effects | UNDRR (2017) |
| Multi-hazard risk | Risk generated from multiple hazards and the interrelationships between these hazards (but not considering interrelationships on the vulnerability level) | Zschau (2017) |
| Multi-risk | Risk generated from multiple hazards and the interrelationships between these hazards (and considering interrelationships on the vulnerability level) | Zschau (2017) |
| Multi-(hazard)-risk | Used in this paper when collectively referring to all of the above | N/A |

*It would be helpful if the challenges listed in section 3 and the suggested research agenda discussed in section 4 would be closer linked and directly referenced in the text. Is there a correspondence between those sections that deserves to be further highlighted?*

Many thanks for this very useful suggestion! Indeed, it is the idea that the research agenda (section 4) links to the challenges raised in section 3. However, based on the reviewer's feedback, we realise that this needs to be made much more explicit. To do this, we have adjusted the ordering of the challenges in section 3 so that they align with the aspects of the research agenda in section 4. In doing so, we have also ensured that the intro to each aspect (i.e. the bold parts) now use the similar text and phrasing. The table below shows these new headings for sections 3 and 4, which now follow the same order. This left a challenge from the original manuscript regarding the part that had the heading "There is a lack of decision makers/institutions dealing with interconnected multi-risks across sectors, hazards and regions"): this part was more overarching, and upon review somewhat repetitive of the text earlier in section 3. Therefore, in the new manuscript, we have integrated it into the introductory part of section 3, to read: "The aforementioned challenges exist within the context of an increasingly interconnected world, increased pressure for space, and climate change, in which the magnitude and frequency of single and multi-hazards are changing at an unprecedented rate (Vogel et al., 2019). Transboundary risk assessment and management for multi-(hazard)-risk across countries (e.g. the 2002 flood events in Eastern Europe or the heat wave in 2003) is still lacking but very much needed in this increasingly interconnected world. As

has been seen with the financial crisis in 2007/08, even in the case that individuals behave rationally in their own way (e.g. making profit) the systemic risk that they were creating together led to the near collapse of the system. The interconnectedness of hazards and risks need to be explicitly taken into account, both from the individual and system (e.g. country/transboundary) perspective, may it be in relation to the economic or social or ecological system at hand, so that individual DRM measures do not produce systemic risks on other spatial scales/sectors in the future."

| Section 3 | Section 4 |
|---|---|
| There is a diverse language on multi-(hazard)-risk and a lack of overview of existing methods and tools | Establishing a set of common multi-(hazard)-risk definitions and concepts, and an overview of existing methods and tools |
| We lack a clear framework and guidelines for multi-(hazard)-risk assessment and management | Co-developing a framework for multi-(hazard)-risk assessment and management |
| Dynamic feedbacks between hazard, exposure, and vulnerability are poorly understood | Increasing understanding of dynamic feedbacks between hazard, exposure, and vulnerability |
| Many past multi-(hazard)-risk projects and accompanying software focus on multiple single hazards under current conditions and tend not to focus multi-(hazard)-risk interactions or future scenarios | Developing future scenarios of plausible multi-(hazard)-risk |
| Only few studies assess the effectiveness of DRM measures across hazards, sectors, and time horizons | Assessing the effectiveness of DRM measures across hazards, sectors, and time horizons |
| There is a distinct lack of in-depth case-studies on multi-(hazard)-risk | Testing of approaches in in-depth case studies on multi-(hazard)-risk assessment and management |

*l.64: Is globalisation and concentration the only driver of increased impacts? Some Natural Hazards surely have become more impactful by increased frequency and magnitude due to Climate Change.*

Indeed, we did not want to imply that this is the only driver, and thanks for pointing out the omission. We added: "Moreover, the intensity and/or frequency of many climate-related hazards is projected to increase in the 21st century (IPCC, 2022)."

Ref: IPCC, 2022. Climate Change 2022. Impacts, Adaptation and Vulnerability. Summary for Policymakers. Working Group II Contribution to the Sixth Assessment Report of the Intergovernmental Panel on Climate Change. Cambridge University Press, Cambridge

*l.153. what would be a 'tool' and what would be a 'method' in this context? Are they the same?*

Thanks for the good question. Within the MYRIAD-EU project, this exact question also arose during the initial review of concepts and terminology. Within MYRIAD-EU, we define a method as a way, technique, or process of/for doing something, and a tool as a resource to help you meet an objective or to generate new knowledge or information. We added these definitions to the manuscript.

*l.145 paradigm shift is needed. For what exactly? For disaster risk management?*

Correct, we have added "…in disaster risk management…"

*The challenge in line l.194 is a bit difficult to understand as the aspects listed in the title are not necessarily related?*

We have re-written and expanded this challenge to increase clarity, to read: "Many past multi-(hazard)-risk projects and accompanying software focus on multiple single hazards under current conditions, and tend not to focus multi-(hazard)-risk interactions or future scenarios"

*l.212 'has been'*

Thanks - added

*l.239 The meaning of this sentence is hard to understand 'We present a research agenda that help us move towards this approach.' Is it rather a research agenda that implements this approach? How do you move towards an approach?*

The formulation was unclear as to what the 'approach' was referring to. We have made this clearer, stating that: "We present a research agenda to help us move towards this approach in which multi-(hazard)-risk management is addressed through the lens of sustainability challenges that cut across sectors, regions, and hazards".

*l.260 what is a 'risk-driver'?*

By 'risk drivers', we meant to refer collectively to hazard, exposure, and vulnerability. However, as 'risk driver' may have a different meaning to different people, we changed to explicitly state 'hazard, exposure, and vulnerability'.

*l.281 This is a bit of a complicated sentence as 'framework' are used twice with likely different meaning? '..achieve this by co-developing the 'framework', and products and services to operationalise the 'framework'.*

The intended meaning of the use of framework is the same in the two cases. We will make this sentence clearly, to read: "The MYRIAD-EU approach aims to achieve this by co-developing the framework, and products and services that can be used to operationalise the framework, with stakeholders in five multi-scale Pilots: North Sea, Canary Islands, Scandinavia, Danube, and Veneto"

*l.297 scales stand for temporal, spatial scales?*

We have clarified that this refers to spatial scales.